# Uncovering common disease mechanisms and critical biomarkers in Crohn's disease with concurrent psoriasis and exploring potential therapeutic agents

Tianqi Liu[1]☯, Xiaoqing Zhang[2]☯, Ruiqi Chen[1], Yifan Sun[1], Ruijian Zhang[1], Liwen Zhang[1], Zhepeng Luo[1], Jiani Wang 🔟[1]*

1 Department of Gastroenterology, First Affiliated Hospital, China Medical University, Shenyang, Liaoning Province, China, 2 Teaching Center for Medical Experiment, China Medical University, Shenyang, Liaoning Province, China

☯ These authors contributed equally to this work.
* jiani.wang@cmu.edu.cn

## Abstract

### Introduction

Research findings show a substantial correlation between Crohn's disease and psoriasis. However, the exact cause or pathogenesis of the concurrent manifestations of these two conditions in the same individuals remains uncertain. This research aimed to scrutinize the important molecules and mechanisms responsible for the concomitance of Crohn's disease and Psoriasis by using quantitative bioinformatics utilizing a publicly available RNA sequencing repository.

### Methods

The database Gene Expression Omnibus were assessed, specifically for Crohn's disease (GSE95095) and psoriasis (GSE13355). The 'limma' library of the R programming syntax is employed to identify differentially expressed genes. The Search Tool for Interacting Genes dataset was utilized to study the interaction between proteins networks. The Cytoscape software was utilized to efficiently view and analyse these Protein-Protein Interaction networks. The ctoHubba Cytoscape plugin helps in the selection of hub genes. These hub genes have been confirmed using data from GSE102133 for Crohn's disease and GSE14905 for psoriasis. The ROC curves were utilized in this study to assess the diagnostic value of the hub genes. Moreover, new research involving gene-set enriched studies and the study of immunological surveillance associated with these specific genes is attainable.

**Data availability statement:** This study utilized the following publicly available gene expression datasets: GSE95095, GSE13355, GSE102133, and GSE14905. These datasets can be queried and downloaded from the Gene Expression Omnibus (GEO) database (https://www.ncbi.nlm.nih.gov/geo/) using the accession numbers mentioned above. Specifically, the access links for dataset GSE95095 are https://www.ncbi.nlm.nih.gov/geo/query/acc.cgi?acc=GSE95095, GSE13355 is https://www.ncbi.nlm.nih.gov/geo/query/acc.cgi?acc=GSE13355, GSE102133 is https://www.ncbi.nlm.nih.gov/geo/query/acc.cgi?acc=GSE102133, and GSE14905 is https://www.ncbi.nlm.nih.gov/geo/query/acc.cgi?acc=GSE14095.

**Funding:** This work was supported by National Natural Science Foundation of China (82000528) and the Natural Science Foundation of Liaoning Province (2021-BS-092). These funds were received by Dr. Jiani Wang.

**Competing interests:** The authors have declared that no competing interests exist.

6 **Abbreviations:** CD, Crohn's disease; DEGs, differentially expressed genes; GEO, Gene Expression Omnibus; KEGG, Kyoto Encyclopedia of Genes and Genomes; PPI, Protein-Protein Interaction Networks.

## Results

Among the identified common DEGs, 40 genes were downregulated and 37 were upregulated, totaling 77 genes. Crohn's disease and Psoriasis had a higher concentration of pathways associated with inflammation. After validation, functionality of hub genes was confirmed for S100A12, CXCL8, IL1RN, S100A9, CXCL10, MMP1, CXCL1, FPR1, CXCR2, and S100A8. The hub genes showed an increase in expression in response to neutrophil infiltration. The expression of S100A12, CXCL8, IL1RN, S100A9, CXCL10, MMP1, CXCL1, FPR1, CXCR2, and S100A8 was found to be significantly linked to immune processes such as neutrophil activation, neutrophil chemotaxis, and neutrophil migration associated with Crohn's and Psoriasis disease.

## Conclusions

This bioinformatics study has elucidated S100A12, CXCL8, IL1RN, S100A9, CXCL10, MMP1, CXCL1, FPR1, CXCR2, and S100A8 as the central genes in the pathogenesis of CD and Psoriasis comorbidity. The significance of neutrophil infiltration in promoting inflammatory and immune-mediated dysfunction seems to be crucial in the etiology of concurrent Crohn's and Psoriasis, offering an avenue for diagnostic and therapeutic methods.

## 1 Introduction

Crohn's disease, an inflammatory granulomatous disorder classified as an inflammatory bowel disease, typically localizes in the terminal ileum and adjacent colon. Despite extensive research, its etiology and pathogenesis remain enigmatic, likely stemming from a combination of genetic, immunological, environmental, and perturbations in gut microbial communities. The typical symptoms are abdominal pain, diarrhea, and mucopurulent stools, which can be recurrent and persistent, severely affecting the patient's quality of life. There is a lack of effective and reliable treatment methods [1].

Psoriasis is an intricate, polygenic hereditary disease caused by the immune system and environmental factors. A scaly red macule or plaque with lamellar scales that can be scraped away to reveal a pale crimson translucent film and peeled away to reveal punctate bleeding is the typical clinical appearance. Lesions can be localized or widespread, are non-infectious, difficult to cure, and can have a substantial impact on patients' health and way of lifestyle [2].

According to research, the prevalence of psoriasis among people with Crohn's disease sufferers is about 9.6%, in contrast to the general population's rate of 2.2 [3]. Whereas, patients with psoriasis are 1.70 times more likely to develop CD [4]. Individuals afflicted with Crohn's disease face an elevated susceptibility to psoriasis, and psoriasis amplifies the susceptibility to developing Crohn's disease. What factors contribute to the correlation between Crohn's disease and psoriasis? The specific link between them in terms of pathogenetics remains unknown. The exact pathogenetic

association between them is not known. In their comprehensive review, C.R.H. Hedin et al. obtained an extensive variety of data and proposed that an immune response within afflicted tissues is caused by a multifaceted interaction of inherited, epigenetic, and biologic factors [5]. Surprisingly, just a few research have looked into shared biochemical markers between the two disorders. The shared transcription trait can shed new light on the common etiology of Crohn's disease and psoriasis. The present study was achieved to uncover the genes linked to the overlap of Crohn's disease and Psoriasis, the researchers used the database known as the Gene Expression Omnibus (GEO) to access tissue expression data from people with Crohn's disease and psoriasis. These profiles were pooled and examined in order to identify shared genetic features or patterns that may shed light on the development of both illnesses when they coexist. This entailed the rigorous application of comprehensive bioinformatics techniques, encompassing enrichment analysis, to reveal common DEGs and elucidate their functional roles in both diseases. To further probe the the interaction between proteins, PPI framework was meticulously developed using the STRING database and Cytoscape software, ultimately enabling the exploration of gene modules and the identification of central hub genes. Ultimately, we discovered a total of ten hub genes, namely S100A12, CXCL8, IL1RN, S100A9, CXCL10, MMP1, CXCL1, FPR1, CXCR2, and S100A8. Our investigation delved deeply into the key mechanisms governed by these genes while confirming their expression across external datasets. The curve of receiver operating characteristics (ROC) is being utilized in a complete analysis to examine the accuracy of the key indicators in their capacity to correctly identify and distinguish between the two diseases, providing an in-depth review of their diagnostic value. We are certain that the hub genes and diagnostic model developed in the context of Crohn's disease and psoriasis will provide unique insights into the intricate processes behind the recurrence of both conditions.

## 2 Methods

### 2.1 Data source

We used the query terms "Crohn's disease" and "Psoriasis" to search the GEO database for relevant genome-wide expression datasets. The microarray information GSE95095 and GSE13355 are taken from the Gene Expression Omnibus (GEO) repository, which may be found at http://www.ncbi.nlm.nih.gov/geo. The GSE95095 data contains information from 24 Crohn's disease patients and twelve healthy control subjects. Within the GSE13355 dataset, there are 58 individuals diagnosed with Psoriasis and 64 individuals who are in a state of normal health. Furthermore, the GEO database provided the GSE102133 (CD) and GSE14905 (Psoriasis) validation datasets.

### 2.2 Identification of genes with differential expression

The statistical R package (GEOquery) was used to access raw information [6], while a method (normalize between arrays) in the R package (limma) serves to eliminate any variances between the samples [7]. Our study uses "Limma" method to analyze the expression of hub genes of the experimental versus control groups in order to find the differentially expressed genes (DEGs). Probe sets lacking a corresponding gene symbol were eradicated. The average value was calculated for genes that had multiple probe sets. Fold changes (FCs) were computed for the expression of each gene. DEGs were defined as having a P-value below 0.05 and a log-fold change ($|logFC|$) that exceeds 1.0. The Venn diagram tool (http://bioinformatics.psb.ugent.be/webtools/Venn) utilized to determine associated DEGs.

### 2.3 Gene enrichment study of gene expression variance

Genome Ontology (GO) is a well-known universal method for recognizing cellular processes, using a precise lexicon and highly specified concepts to find genes and their associated genes. The GO analysis comprised of 3 ontologies including functional molecules, cellular elements, and biological mechanisms. When compared to the genetic baseline, the GO enrichment algorithm produces a list of GO terms that are enriched for DEGs, while simultaneously filtering out DEGs

associated with biological functions [8]. The Kyoto Encyclopedia of Genes and Genomes (KEGG) stands as the primary repository of information pertaining to public pathways. Pathway enrichment analysis is used to identify pathways exhibiting sufficient DEG enrichment relative to the genome-wide backdrop [9]. The "clusterProfiler" R package employed to carry out GO and KEGG enrichment analyses [10].

## 2.4 Module analysis and establishment of the Protein-Protein Interaction (PPI) analysis

In PPI evaluation, we used STRING (http://string-db.org), which is a web-based tool, to investigate interactions between genes with a total value greater than 0.4 [11]. This allowed us to uncover associations such as direct binding connections or shared regulatory pathways, ultimately creating a comprehensive protein-protein interaction network with numerous regulatory relationships. To visualize this network, we used Cytoscape software, a powerful tool for network analysis. We used Cytoscape's molecular complex identification (MCODE) approach to find the important functional regions in the entire structure [12]. A K-core value of 2, a degree of threshold of 2, a maximum level of 100, and a node value limit of 0.2 were among the criteria utilized to identify these modules. These criteria were used to separate and extract important modules or subnetworks from a larger network or dataset.

## 2.5 Selecting and evaluating hub genes

When it came to selecting and analyzing hub genes, we used Cytoscape's cytoHubba plug-in, which offers various algorithms for this purpose [13]. To assess and identify the hub genes in our network, we utilized the following algorithms: MCC, MNC, DMNC, Degree, and EPC. At long last, the genes identified by the five algorithms were determined to be dependable hub genes and illustrated with a Venn diagram. Using this comprehensive approach, we were able to identify genes which contribute a significant part in the coexistence of Crohn's syndrome and psoriasis.

## 2.6 Construction of Receiver Operating Characteristic (ROC) curves

The identified hub genes' expression of mRNA was validated in both GSE102133 and GSE14905. Under the GSE102133 database, there were 12 controls and 65 individuals diagnosed with CD. Within GSE14905, there were 21 controls and 33 Psoriasis samples. We employed Graphpad Prism to create the ROC curves and then worked out the AUC of the hub genes, which showed their diagnostic effectiveness. Statistical significance was attributed to p-values below 0.05 and AUC values surpassing 0.7. We obtained valuable information about the significant hub genes by utilizing Human Protein Atlas server, accessible at https://www.proteinatlas.org

## 2.7 Hub gene network analysis

The GeneMANIA platform, which is available at http://www.genemania.org, was significant in establishing an integrated expression network for the hub genes. This approach is widely used for revealing the complicated interconnections and interactions across gene sets, hence improving our understanding of the functional connections of the hub genes [14].

## 2.8 Immune Infiltration and the association of Hub Gene

To estimate the number of immunological cells, we used the GSVA package in R software, which relied on the reference expression of genes within the gene set. Our research included 28 different cells. Present study employed the single-sample gene set enrichment analysis (ssGSEA) approach to calculate immunological ratings for individual samples. This method allows us to assess the immunological characteristics of each sample, providing insights into the immune-related aspects and potential differences among the samples under investigation. This method aided us in determining the distribution of immunological cells in the samples, offering useful insights into the immunological milieu of the conditions under study [15].

## 2.9 Enrichment analyses of hub genes

The following approaches were used: GSEA (Gene Set Enrichment Analysis) and GO (Gene Ontology) enrichment studies. The preceding section described the GO enrichment analysis method. We studied variations in gene expression based on demographic traits for GSEA, with the aim of evaluating the interrelated pathways and biological mechanisms between the two groups under examination. GSEA is a powerful tool for detecting coordinated changes in gene sets that can provide important insights into the underlying biology and functional differences between these groups [16]. In our analysis, significant genes met the following criteria: nominal P values less than 0.05, |normalized enrichment scores (NES)| more than 1, and a false discovery rate (FDR) q value less than 0.25. Only gene sets with strong statistical significance and considerable enrichment were considered significant.

## 2.10 Discovery of transcription factors and miRNAs interaction

Using the TRRUST database, we studied the relationship between hub genes and hub transcription factors (TFs) [17], Additionally, we identified hub miRNAs that negatively influenced protein expression by binding to hub gene transcripts, a discovery made through mirTarbase [18]. We used Cytoscope to build a network of TFs-gene and miRNA-gene interactions.

## 2.11 Analysis of gene-drug interactions

To uncover potential drugs interplaying with the hub-genes, we consulted the drug-gene interaction database (https://dgidb.genome.wustl.edu/) [19]. In addition, we utilized the SymMap database to investigate possible interaction of traditional Chinese medicines with hub-genes [20].

## 2.12 Statistical analysis

The data obtained for the current study was extensively analyzed using R software and GraphPad Prism. Statistical significance was evaluated at a significance level of $P < 0.05$. This strict criterion guaranteed that only results with a high level of confidence were considered significant.

## 3 Results

### 3.1 Screening of common differential Genes

The basic information from the datasets related to Crohn's disease and Psoriasis is shown in Table 1. With P value < 0.05 and |log2FC| > 0.5 as the screening conditions, a total of 715 genes were up regulated and 1578 genes were down regulated in Crohn's disease(Fig 1A). In Psoriasis, a total of 658 genes remained up regulated and 346 genes were down

**Table 1. Information of selected four datasets.**

| Group | GSE number | Platform | Disease | Samples | Tissue type |
|---|---|---|---|---|---|
| Exploration cohort | GSE95095 | GPL14951 | Crohn's disease | Normal: 12 CD: 24 | ileum |
| | GSE13355 | GPL570 | Psoriasis | Normal: 64 Psoriasis: 58 | skin |
| Validation cohort | GSE102133 | GPL6244 | Crohn's disease | Normal: 12 CD: 65 | ileum |
| | GSE14905 | GPL570 | Psoriasis | Normal: 21 Psoriasis: 33 | skin |

regulated(Fig 1B). In total, 77 common DEGs were detected following the intersection of the Venn diagrams. Among these, 40 were found to be downregulated, while 37 were upregulated (Fig 1C, Fig 1D). Table 2 contains detailed information on the co-DEGs.

### 3.2 Evaluation of the shared pathways of co-DEGs

For the assessment of the functional roles and pathways associated with the 77 shared DEGs between Crohn's disease (CD) and Psoriasis, we employed the "clusterprofiler" tool. The results, as depicted in Fig 2A, revealed enriched biological processes and molecular functions, although no significant enrichment in cellular components was observed. Notably, the molecular function analysis revealed that genes involved in activities such as leukocyte chemotaxis, cell movement, neutrophil movement, granulated chemotaxis, migratory neutrophil granulocyte migration, and myeloid proliferation were over-represented. The results of the biological process showed that these genes had increased binding to the RAGE receptor and increased cytokine activity.Furthermore, a KEGG pathway enrichment analysis revealed that IL-17 signaling pathway and cytokine-cytokine receptor interaction were heavily enriched (Fig 2B). The findings show that inflammatory mediators have a significant part in the progression of both of these disorders.

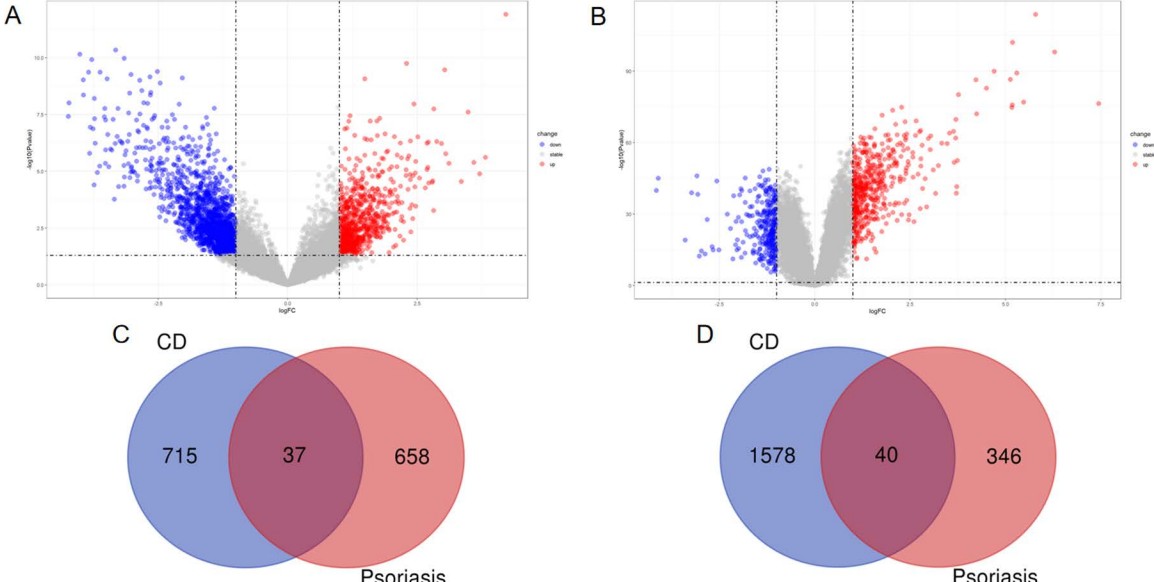

**Fig 1. Identification of differentially expressed genes.** (A) The volcano map of GSE95095. (B) The volcano map of GSE13355; upregulated genes are marked in red; downregulated genes are marked in blue. (C) The two datasets showed an overlap of 37 upregulated DEGs. (D) The two datasets showed an overlap of 40 downregulated DEGs.

**Table 2. Common Differential Genes.**

| Change | Symbol |
|---|---|
| Up | NAMPT, S100A9, FOSL1, MMP9, CYP7B1, MMP1, SELL, CXCL1, PRSS2, IL1RN, CXCR2, LTB, CFB, CCL19, SQLE, ODF3B, FPR1, SOCS3, CXCL8, EPHX3, COMP, CXCL10, IDO1, VNN1, S100A12, STEAP4, IFI16, TYMP, GBP1, CEMIP, S100A8, SOX7, GZMA, SPRR1B, MOXD1, CRABP2, MICALL1 |
| Down | HMGCS2, NR3C2, CYP2J2, CDHR1, CLDN8, PLEKHH1, CLDN23, AGR3, RHPN2, SLITRK6, PPARGC1A, TLCD4, SCARA5, SLC26A2, PRLR, TSPAN8, FA2H, ALDH3A2, RNASE4, TMEM97, RETREG1, CAMK2N1, ZNF91, TCF7L2, SOX6, EFNB2, DDAH1, SCGB2A1, CHP2, EEF2K, EPCAM, CHL1, FCGBP, SYBU, LGR5, HLA-DQB2, BHLHE41, ACOX2, ENPP5, CGNL1 |

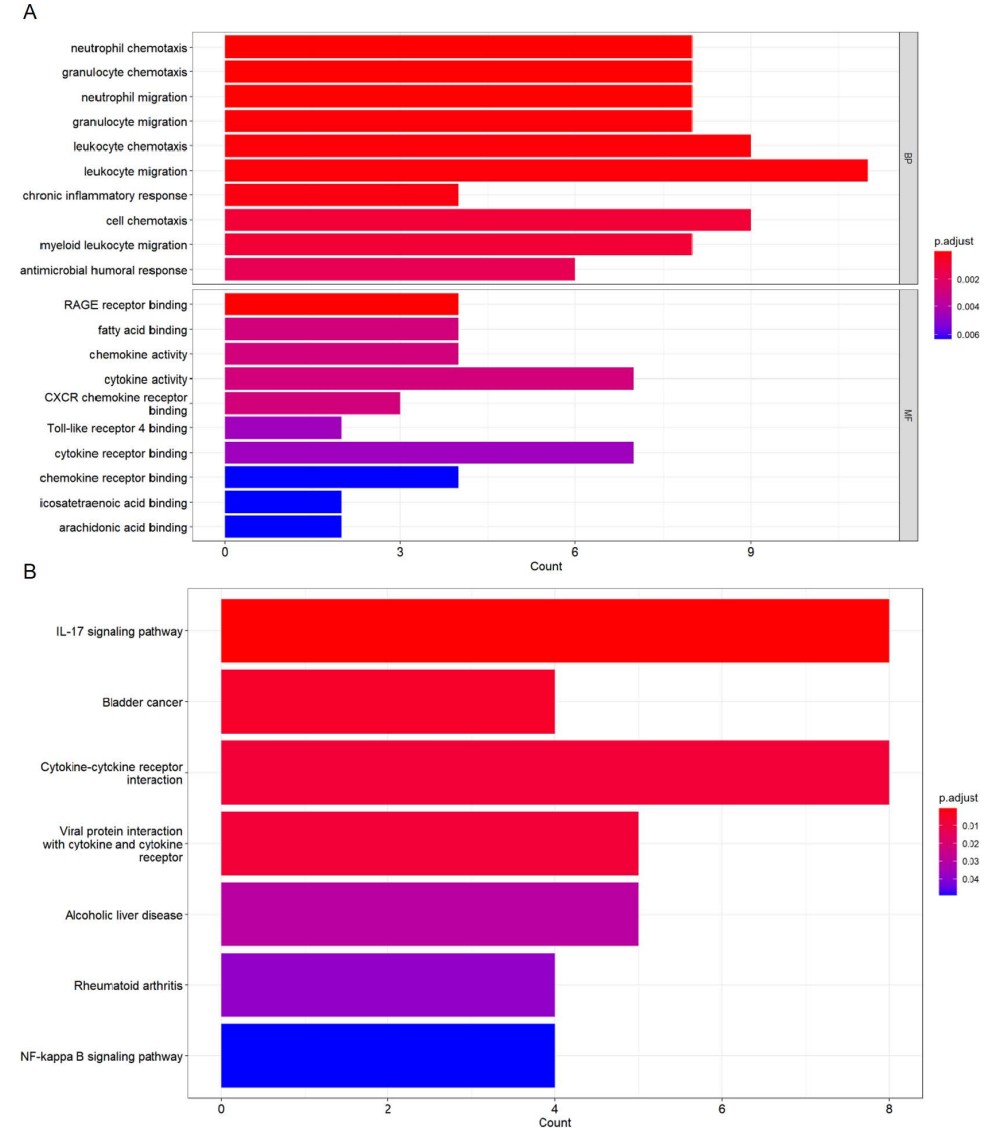

**Fig 2. The enrichment analysis results of co-DEGs.** The enrichment analysis results of GO (A) and KEGG (B) pathway.

### 3.3 Building the PPI network for co-DEGs

With the database of STRING, we performed a PPI-based network analyses for the DEGs that comply with both conditions, as shown in Fig 3A. This analysis allowed us to explore the interactions among these DEGs. The STRING files were imported into Cytoscape software to generate protein interaction networks, with higher scores resulting in darker circles and larger areas (Fig 3B). In the MCODE plug-in, the K-core value > 2 was set to obtain 2 clusters, cluster1 (Fig 3C) (score: 5.75) including CXCL1, FPR1, MMP9, SOCS3, MMP1, CXCR2, CXCL10, SELL and CCL19, of which the seed gene is CXCL1; cluster2 (Fig 3D) (score: 4) includes S100A8, S100A9, S100A12 and CXCL8, of which the seed gene is S100A8.

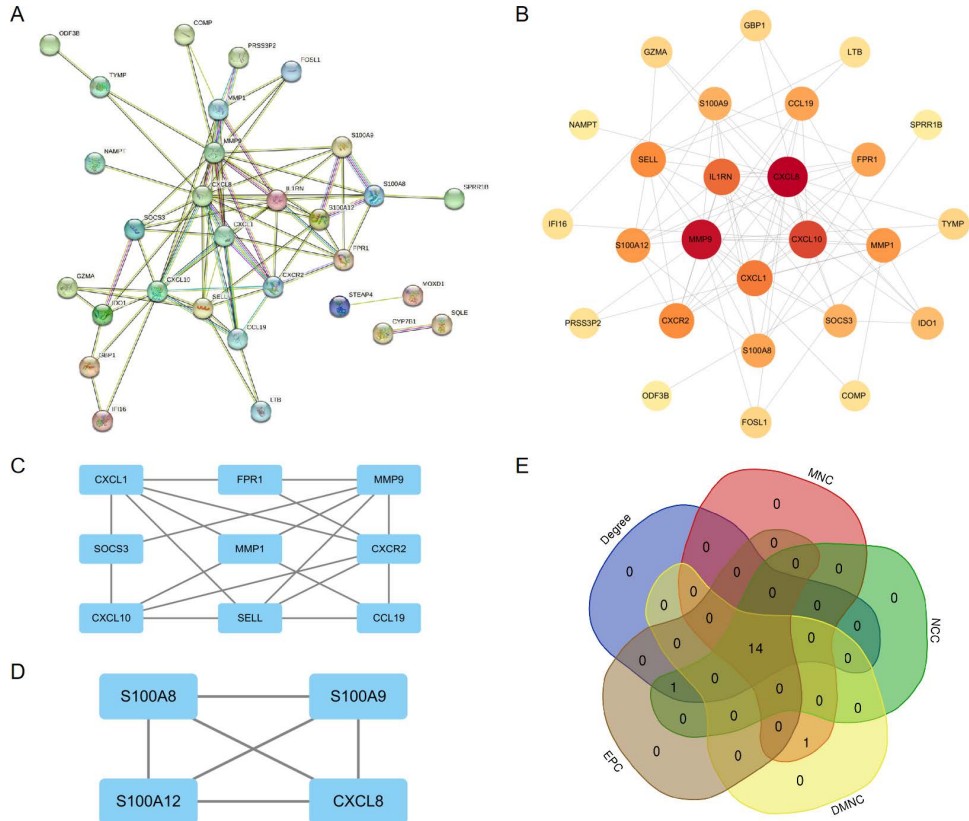

**Fig 3. the PPI Network for co-DEGs.** (A) PPI network of the common DEGs constructed by STRING. (B) PPI network of the DEGs constructed by Cytoscape. (C,D) Two gene clustering modules constructed by the MCODE plug-in. (E) Identification of 14 hub genes for hub genes by five algorithms.

## 3.4  Identification and verification of hub gene

Using the cytoHubba plugin, we applied five algorithms (MCC, MNC, DMNC, Degree, and EPC) to detect the top 15 hub genes among the DEGs publicly accessible(Table 3). The Venn diagrams revealed that the five algorithms produced a combined total of 14 hub genes, which included S100A12, SELL, SOCS3, CXCL8, IL1RN, CCL19, S100A9, CXCL10, MMP1, MMP9, CXCL1, FPR1, CXCR2 and S100A8 (Fig 3E). Then, we constructed ROC curves to validate the diagnostic value of the 14 hub genes obtained from the PPI network on the common DEGs based on the GSE102133 and GSE14905 datasets (Fig 4). The AUC and P values of each hub gene are shown in Table 4. Diagnostic accuracy was achieved when the AUC was greater than 0.7 and the P value was less than 0.05. According to this diagnostic criterion, we considered ten genes such as S100A12, CXCL8, IL1RN, S100A9, CXCL10, MMP1, CXCL1, FPR1, CXCR2, and S100A8 as having some diagnostic effect. The details of the 10 hub genes with diagnostic significance that we obtained in the human protein atlas are shown in Table 5.

## 3.5  PPI Construction of hub genes

In terms of co-localization, typical structural categories, expression, forecasting, and networks, GeneMANIA investigates the interplay and relationship that exists among the 10 hub genes and their associated genes. Fig 5A shows two concentric circles, each representing a different set of genes. The expected genes are on the outer circle, while the hub genes are in the inner circle. This diagram depicts the overabundance of these genes in a variety of pro-inflammatory processes.

**Table 3. Top 15 hub genes in five algorithms.**

| Algorithms | MCC | MNC | DMNC | Degree | EPC |
|---|---|---|---|---|---|
| Top 15 hub genes | CXCL8<br>MMP9<br>CXCL1<br>CXCL10<br>CXCR2<br>SELL<br>CCL19<br>IL1RN<br>S100A12<br>S100A8<br>S100A9<br>FPR1<br>MMP1<br>SOCS3<br>IDO1 | CXCL8<br>MMP9<br>CXCL10<br>IL1RN<br>CXCL1<br>SELL<br>CXCR2<br>MMP1<br>S100A12<br>FPR1<br>CCL19<br>S100A8<br>SOCS3<br>S100A9<br>IDO1 | S100A8<br>S100A9<br>CXCR2<br>S100A12<br>CXCL1<br>FPR1<br>CCL19<br>IL1RN<br>SOCS3<br>SELL<br>FOSL1<br>CXCL8<br>CXCL10<br>MMP1<br>MMP9 | CXCL8<br>MMP9<br>CXCL10<br>IL1RN<br>CXCL1<br>SELL<br>CXCR2<br>MMP1<br>S100A12<br>S100A8<br>FPR1<br>CCL19<br>SOCS3<br>S100A9<br>IDO1 | CXCL8<br>MMP9<br>CXCL10<br>CXCL1<br>IL1RN<br>CXCR2<br>SELL<br>S100A12<br>FPR1<br>CCL19<br>MMP1<br>SOCS3<br>S100A9<br>S100A8<br>IDO1 |
| Fourteen overlapping hub genes | S100A12,SELL,SOCS3,CXCL8,IL1RN,CCL19,S100A9,CXCL10,MMP1,MMP9,CXCL1,FPR1,CXCR2,S100A8 | | | | |

## 3.6 Association between the hub genes and immune infiltration

Building upon the findings from the GeneMANIA examination, we utilized Spearman's correlation analysis to explore the connection between the hub genes and immune cells. Fig 5B illustrates that the infiltration level of Gamma delta T cell, Type 17 T helper cell, Regulatory T cell and CD56dim natural killer cell was significantly linked to S100A12, CXCL8, IL1RN, S100A9, CXCL10, MMP1, CXCL1, FPR1, CXCR2, and S100A8 in CD samples of GSE102133. The presence of S100A12, CXCL8, IL1RN, S100A9, CXCL10, MMP1, CXCL1, FPR1, CXCR2, and S100A8 proved to be significantly interrelated with the infiltration level of Activated dendritic cell, Macrophage, and Neutrophil in Psoriasis samples in GSE111889 (Fig 5C).

## 3.7 Hub gene enrichment evaluation

By employing GO and GSEA to do an enrichment analysis on the 10 hub genes, and the results revealed that hub genes are mostly important for neutrophil, granulocyte, granulocyte, granulocyte, and leukocyte chemotaxis (Fig 6A). These findings highlighted the significance of the inflammatory response in these two diseases. In the meantime, we used GSEA to do a KEGG analysis of hub genes. Following the Gene Set Enrichment analysis, hub genes were found to be heavily connected with inflammation-related pathways (Fig 6B, 6C). The GO enrichment study of 10 hub genes agrees with the GSEA enrichment analysis, adding support to the concept that inflammatory pathways represent the common pathogenic mechanism of CD and Psoriasis.

## 3.8 Construction of TF-miRNA regulatory networks

In the analysis of hub genes, TRRUST and mirTarbase unveiled a total of 18 transcription factors and 20 miRNAs found within the gene-TF (Fig 7A) and gene-miRNAs (Fig 7B) interaction networks.

## 3.9 Drug prediction using hub genes

The ten hub genes were added to the DGIdb database, the Preset Filters default values were used, the three screening terms Approved, Antineoplastic, and Immunotherapies were checked, the reliability of the evidence of gene-drug interactions was suggested by the interaction scores in the exported results, and the 8 species with scores ≥ 0.1 were screened

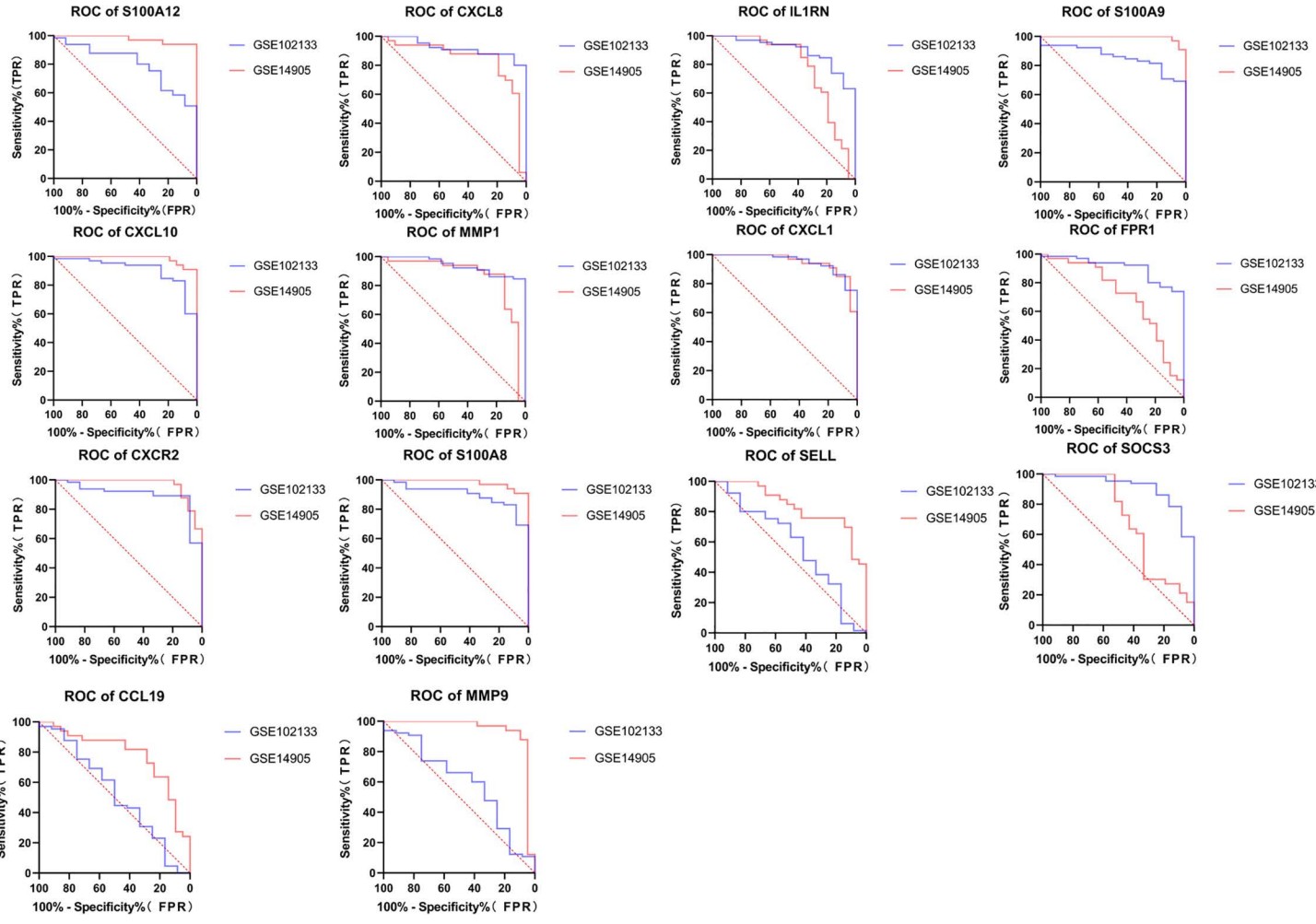

**Fig 4. Validation of hub genes in the diagnostic value.**

drugs (Table 6). Among them, Methotrexate could target S100A8, IL1RN and S100A12 simultaneously. Next, each of the 10 hub genes was entered into the SymMap database, and target-associated TCM was derived and counted using a Venn diagram.The results show that there are 21 drugs that can treat more than seven target genes and have been shown to treat two diseases (Table 7).

## 4 Discussion

In this study, we examined the GSE95095, GSE13355, GSE102133, and GSE14905 datasets retrieved from GEO to identify DEGs in patients experiencing CD and Psoriasis. Following that, KEGG and GO pathway enrichment analyses were done to acquire a mechanistic understanding of the activities and pathways linked with these DEGs. We used Cytoscape to perform a comprehensive bioinformatics analysis to identify S100A12, CXCL8, IL1RN, S100A9, CXCL10, MMP1, CXCL1, FPR1, CXCR2 and S100A8 as hub genes.

S100 proteins are a subfamily of the calcium-binding protein family. s100A8, s100A9 and s100A12 are members of the s100 protein family. s100A8, s100A9 and s100A12 exist as homodimers and are similar in structure and function. S100A8

**Table 4. The area under the curve (AUC) and P-value of the fourteen hub genes.**

| Hub gene | GSE102133 | | GSE14905 | |
|---|---|---|---|---|
| | AUC | P | AUC | P |
| S100A12 | 0.8026 | 0.0009 | 0.9784 | <0.0001 |
| SELL | 0.5744 | 0.4153 | 0.8341 | <0.0001 |
| S0CS3. | 0.9128 | <0.0001 | 0.6955 | 0.0162 |
| CXCL8 | 0.9192 | <0.0001 | 0.8427 | <0.0001 |
| IL1RN | 0.8974 | <0.0001 | 0.7821 | 0.0005 |
| CCL19 | 0.5269 | 0.7680 | 0.7835 | 0.0005 |
| S100A9 | 0.8577 | <0.0001 | 0.9942 | <0.0001 |
| CXCL10 | 0.9103 | <0.0001 | 0.9870 | <0.0001 |
| MMP1 | 0.9385 | <0.0001 | 0.8658 | <0.0001 |
| MMP9 | 0.5974 | 0.2858 | 0.9408 | <0.0001 |
| CXCL1 | 0.9513 | <0.0001 | 0.9481 | <0.0001 |
| FPR1 | 0.9077 | <0.0001 | 0.7157 | 0.0080 |
| CXCR2 | 0.9000 | <0.0001 | 0.9668 | <0.0001 |
| S100A8 | 0.9026 | <0.0001 | 0.9827 | <0.0001 |

**Table 5. The detail of ten hub genes from The Human Protein Atlas.**

| Gene name | Protein | Molecular function | Biological process |
|---|---|---|---|
| S100A12 | S100 calcium binding protein A12 | Antibiotic, Antimicrobial, Fungicide | Immunity, Inflammatory response, Innate immunity |
| CXCL8 | C-X-C motif chemokine ligand 8 | Cytokine | Chemotaxis, Inflammatory response |
| IL1RN | Interleukin 1 receptor antagonist | Not found | Not found |
| S100A9 | S100 calcium binding protein A9 | Antimicrobial, Antioxidant | Apoptosis, Autophagy, Chemotaxis, Immunity, Inflammatory response, Innate immunity |
| CXCL10 | C-X-C motif chemokine ligand 10 | Cytokine | Chemotaxis, Inflammatory response |
| CXCL1 | C-X-C motif chemokine ligand 1 | Cytokine, Growth factor | Inflammatory response |
| MMP1 | Matrix metallopeptidase 1 | Hydrolase, Metalloprotease, Protease | Collagen degradation, Host-virus interaction |
| FPR1 | Formyl peptide receptor 1 | G-protein coupled receptor, Receptor, Transducer | Chemotaxis |
| CXCR2 | C-X-C motif chemokine receptor 2 | G-protein coupled receptor, Receptor, Transducer | Chemotaxis |
| S100A8 | S100 calcium binding protein A8 | Antimicrobial | Apoptosis, Autophagy, Chemotaxis, Immunity, Inflammatory response, Innate immunity |

and S100A9 proteins can form homodimers and heterodimers in vivo, and the heterodimer is the most stable form and is the main form in which the proteins exert their biological effects [21–23]. Because S100A8/S100A9 are mostly expressed in myeloid-related cells such as monocytes, neutrophils, and macrophages, they are also known as MRP 8 and 14 [21]. As a heterodimer composed of S100A8 and S100A9 proteins, calmodulin can act as a candidate marker for inflammatory diseases, and can also function to regulate the cytoskeleton and promote leukocyte migration [24]. Several previous studies performed that S100A8/S100A9 can show a pro-inflammatory activity in some autoimmune diseases and these genes were reportedly upregulated in inflammatory states [25,26]. Free S100A8 and S100A9, and S100A8/S100A9 in the extracellular matrix, greatly improve migrating neutrophils to the site of inflammatory processes [27]. The activation and activity of S100A8 and S100A9 is an indicator of psoriasis and may indicate the response of the immune system to psoriasis [28,29]. Furthermore, S100A8 and S100A9 may inhibit psoriasis by prohibiting the synthesis of IL-17A and IL17-F [30].

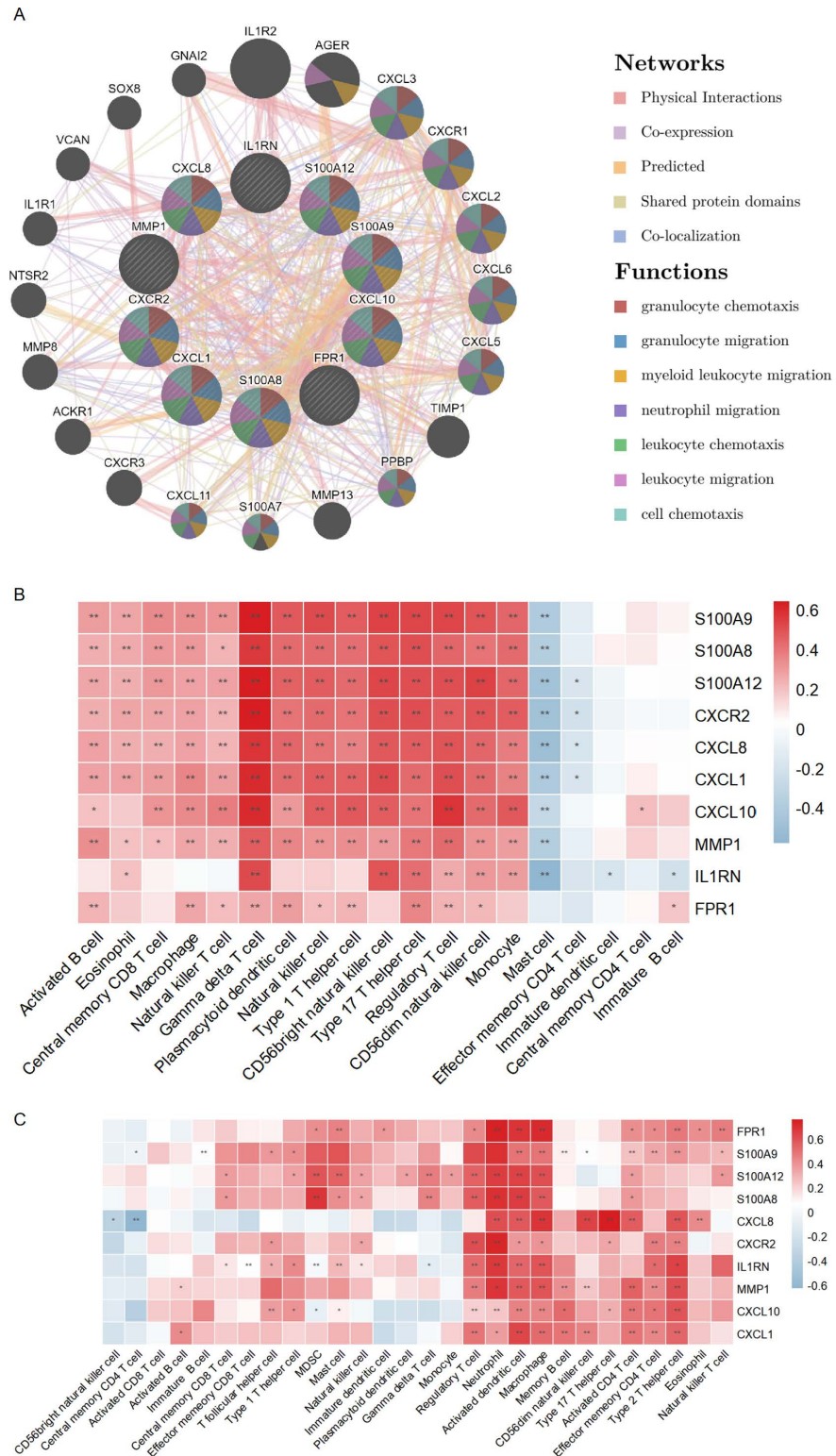

**Fig 5. The PPI and Immune Infiltration of Hub Genes.** (A) Hub genes and their co-expression genes were analyzed via GeneMANIA. (B) Association between the hub genes and immune infiltration in GSE95095. (B) Association between the hub genes and immune infiltration in GSE13355.

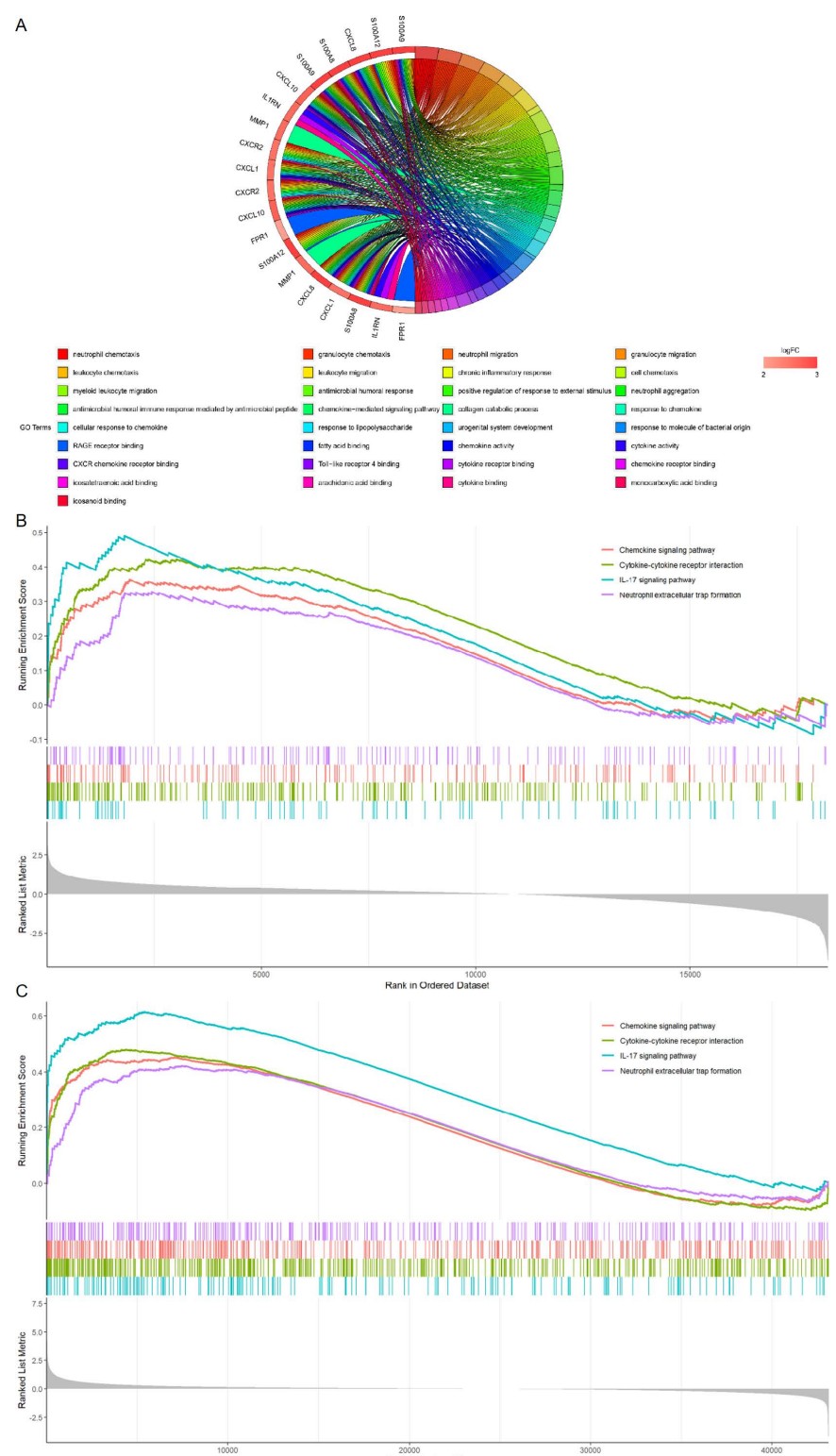

**Fig 6. The enrichment analysis of the hub genes.** (A) GO enrichment analysis of the hub genes. (B) A merged enrichment plot of ten hub genes from gene set enrichment analysis in GSE95095. (C) A merged enrichment plot of ten hub genes from gene set enrichment analysis in GSE13355.

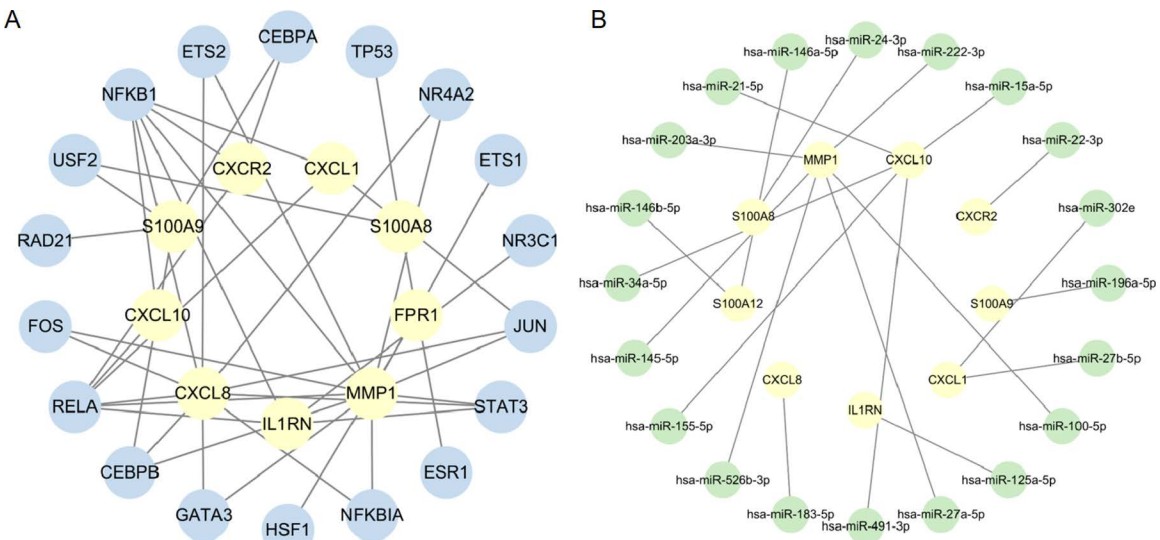

**Fig 7. The interaction networks of gene-TF and gene-miRNAs.** (A) TF-mRNA network of hub genes; (B) miRNA - mRNA network of hub genes.

**Table 6. Prediction results of chemical drugs.**

| Gene symbol | Drug | Interaction score |
|---|---|---|
| S100A8 | METHOTREXATE | 1.14 |
| CXCL10 | METHYLPREDNISOLONE | 0.69 |
| MMP1 | LEFLUNOMIDE | 0.61 |
| IL1RN | METHOTREXATE | 0.38 |
| CXCL10 | OXALIPLATIN | 0.28 |
| S100A12 | METHOTREXATE | 0.23 |
| CXCL8 | LEFLUNOMIDE | 0.18 |
| MMP1 | SIROLIMUS | 0.15 |

A comprehensive transcriptional analysis has identified exclusive regulation of S100A8 and S100A9, both of which are linked to acute inflammatory responses, within the ileal mucosa of CD patients [31]. Inhibiting S100A9 can improve systemic and neuroinflammation driven by local inflammation in the colon [32]. S100A12 protein, also known as calreticulin C, is primarily released by neutrophils and controlling the modulation of inflammatory actions, cell growth, differentiation, and apoptosis [33]. Its main role is to upregulate the level of vascular endothelial cell adhesion molecules, activate inflammatory cells, and exert chemotactic and antimicrobial effects [34]. Previous research has found that S100A12 levels are closely linked to the occurrence of several acute and chronic inflammatory disorders [35,36]. The serum concentrations of S100A12 in both CD and psoriasis patients markedly exceeded optimal levels and exhibited a association with the severity of the disease [37–39]. Research has shown that S100A12 is not expressed in normal skin, but can be seen in the upper basal layer of the epidermis and keratinocytes of hair follicles in plaque psoriasis lesions. By interacting with RAGE as a pro-inflammatory factor, it exacerbates the symptoms of psoriasis [40]. In clinical practice, S100A12 is anticipated to serve as a diagnostic biomarker and aid in assessing disease activity in individuals with Crohn's disease [41].

CXCL8, CXCL10 and CXCL1 belong to chemokine (C-X-C motif) ligand (CXCL) family. CXCL family plays an important role in inflammation [42]. CXCL1, also referred to as Neutrophil Activating Peptide-3 (NAP-3), is named based on its

**Table 7. Prediction results of traditional Chinese medicine.**

| Gene symbol | Latin name | English name |
|---|---|---|
| CXCL1,CXCL10,CXCL8,CXCR2,FPR1,MMP1,S100A12,S100A8,S100A9 | Polygoni Cuspidati Rhizoma Et Radix<br>Choerospondiatis Fructus | rhizome of Gaint Knotweed<br>fruit of Axillary choerospondias |
| CXCL1,CXCL10,CXCL8,MMP1,S100A12,S100A8,S100A9 | Ilicis Cornutae Folium<br>Mori Fructus<br>Mume Fructus<br>Fagopyri Dibotryis Rhizoma | Folium Ilicis Cornutae<br>Mulberry Fruit<br>Smoked Plum<br>Wild Buckwheat Rhizome |
| CXCL10,CXCL8,IL1RN,MMP1,S100A12,S100A8,S100A9 | Astragali Complanati Semen | Flatstem Milkvetch Seed |
| CXCL1,CXCL10,CXCL8,CXCR2,FPR1,MMP1,S100A8 | Astragali Radix<br>Radix Boehmeriae<br>Phellodendri Amurensis Cortex<br>Hyperici Perforati Herba | root of Membranous Milkvetch<br>Ramine root<br>Phellodendron bark<br>Herbahypericiperforati |
| CXCL1,CXCL10,CXCL8,IL1RN,MMP1,S100A8,S100A9 | Armeniacae Semen Amarum | bitter Apricot seed |
| CXCL1,CXCL10,CXCL8,MMP1,S100A12,S100A8 | Salviae Miltiorrhizae Radix Et Rhizoma | root of Ligulilobe sage |
| CXCL1,CXCL8,MMP1,S100A12,S100A8,S100A9 | Cimicifugae Rhizoma | Largetrifolioliious Bugbane Rhizome |
| CXCL10,CXCL8,MMP1,S100A12,S100A8,S100A9 | Rosae Laevigatae Fructus<br>Prinsepiae Nux<br>Hippophae Fructus<br>Nelumbinis Folium<br>Portulacae Herba<br>Moutan Cortex | fruit of Cherokee Rose<br>Hedge Prinsepia Nut<br>fruit of seabuckthorn<br>Lotus leaf<br>all-grass of Purslane<br>Tree Peony Bark |
| CXCL1,CXCL10,CXCL8,CXCR2,MMP1,S100A8 | Catechu | Cutch, Black Catechu |

capacity to stimulate the chemotaxis of neutrophils [43]. CXCL1 is a C-X-C motif chemokine expressed by macrophages, neutrophils and epithelial cells that mediates the inflammatory response by recruiting leukocytes and activating pro-inflammatory mediators [44]. CXCL1 levels are substantially higher in CD patients' inflamed intestinal mucosa than in remission, although CD patients in remission have higher CXCL1 levels in their intestinal mucosa than healthy participants [45]. CXCL1 is also present during the development of psoriasis and can be produced by keratinocytes, fibroblasts, endothelial cells, monocytes, etc. It has chemotactic activity on neutrophils [46]. CXCL8, also called IL8 (interleukin-8), primarily originates from various immune and endothelial cells, and CXCR2 (CXC chemokine receptor 2) is its main functional receptor [47]. CXCL8 is abundantly expressed in pro-inflammatory cells and mediates the patient's inflammatory phase [48]. CXCL8 has two main functions in the body, namely chemoattraction and activation of various types of immune cells. CXCL8 plays a crucial role in psoriasis pathogenesis, particularly in neutrophil infiltration, angiogenesis and keratinocyte proliferation within psoriatic lesions [49]. One study found that chemokine CXCL8 expression was upregulated in mucosal biopsies from patients with Crohn's disease and ulcerative colitis and that this upregulation correlated with disease activity [50]. CXCL8 levels were considerably higher in acute inflammatory bowel conditions and associated with gut inflammation levels [49]. In the colitis mouse model of induced by sodium dextran sulfate (DSS), CXCR2 is highly expressed in inflammatory intestinal tissue [51].In addition to inducing neutrophil activation, CXCR2 can also mediate the activation and proliferation of keratinocytes, which together lead to characteristic changes in the epidermis of psoriatic skin lesions [52]. Owing to the study, the CXCLs/CXCR2 signaling cascade has a critical part in regulating the inflammatory process of inflammation. After inflammation occurs, CXCR2, combined with its ligand, can chemotactic white blood cells and lymphocytes represented by neutrophils to the inflammatory site. When these inflammatory cells moderately aggregate, they can kill pathogenic microorganisms and protect tissues, but when excessively aggregated or activated, they can degrade and damage normal tissues through the release of reactive oxygen species and enzymes, exacerbating the degree of inflammation [53]. Blocking CXCR2 can effectively reduce the aggregation of neutrophils in pneumonia mice and alleviate lung inflammation [54]. Therefore, targeting CXCR2 to regulate neutrophil aggregation is considered an important way

to treat inflammation [55]. CXCL10, also known as interferon-inducible protein-10 (IP-10), has CXCR3 (CXC chemokine receptor 3) as its receptor. CXCL10 is produced by various cell types, including immune cells and non-immune cells, modulating immune responses [56]. Recent studies have shown that CXCL10 expression is increased in the serum and/ or tissues of patients with various autoimmune diseases (e.g., rheumatoid arthritis, systemic lupus erythematosus, desiccation syndrome, systemic sclerosis, and idiopathic inflammatory myopathies). Crypt cell renewal is necessary for normal intestinal homeostasis and mucosal regeneration after injury, and endogenously produced chemokine CXCL10 was found to regulate crypt cell proliferation [57]. Oral administration of sodium dextran sulfate to mice resulted in increased expression of CXCL10 and CXCR3 in the epithelium of the proliferative zone [50]. Neutralization of CXCL10 prevented intestinal epithelial ulceration in mice by promoting the survival of crypt cells [57]. Therefore, targeting CXCL10 may offer a novel treatment strategy for managing inflammatory bowel disease and regulating intraepithelial homeostasis [57]. CXCL10 expression in epidermal keratin-forming cells of psoriatic lesions expression is increased [58]. It has been suggested that CXCL10 may be involved in the pathogenesis of PsA and is considered a predictive biomarker for psoriasis [59].

IL-1Ra, an anti-inflammatory cytokine encoded by the IL1RN gene, is a natural antagonist of the IL-1 receptor, consisting of 152 amino acids with 19% amino acid sequence homology to IL-1α and 21% homology to IL-1β, which competitively binds to the IL-1 receptor [60]. It has been shown that innate IL1RN gene mutations are associated with the development of psoriasis, and animal studies have also found that knocking out the IL1RN gene can contribute to the development of psoriasis [61,62]. IL1-RN levels of IBD patients could potentially be used as predictors of the disease course [63]. However, the mechanism of action of IL1RN in CD remains unexplored. FPR1 is a member of the formyl peptide receptors (FPRs) family, a family of seven transmembrane G protein-coupled chemokine receptors (GPCRs). When activated, FPRs not only strongly mediate chemotactic responses but also promote cell proliferation, differentiation and secretion [64,65]. A curative role of FPR1 has been demonstrated in experimental colitis in mice, but not in human experiments [66]. More and more studies have demonstrated the importance of neutrophils participating in psoriasis. The formation of Munro microabscesses is a significant characteristic, as it is associated with the buildup of neutrophils in the epidermis. The response of neutrophils to chemotactic molecular patterns (PAMPs and DAMPs) triggered by pathogens/injury involves the recognition of these patterns by cell surface transmembrane GPCRs, such as FPR1. Studies have indicated that the alteration of FPR1 function using natural compounds like 3β-hydroxyurs-12,18-dien-28-oic acid (randialic acid B, RAB) and 3β-hydroxyurs-12,19-dien-28-oic acid (tomentosolic acid, TA), has the potential to mitigate in vivo psoriasis-like inflammation [67]. The matrix metalloproteinase (MMP) family is a lineage of metallo-zinc dependent protein hydrolases. These are able to decompose the matrix of cells and play a critical part in cell proliferation and metastasis. The primary substrate targeted is fibrillar collagen, leading to the degradation of collagen fibers and gelatin in the extracellular matrix, thereby inducing changes in the cellular microclimate [68,69]. MMP-1 serum levels are elevated in psoriatic arthritis [70]. In CD, MMPs are involved in tissue remodeling, angiogenesis, and promotion of leukocyte extravasation in areas of active inflammation at the base of the ulcer [71].

Psoriasis and Crohn's disease are both immune mediated chronic recurrent inflammatory diseases. Research has shown that patients with Crohn's disease have an increased risk of developing psoriasis, which also increases the risk of Crohn's disease [72]. Patients with both Crohn's disease and psoriasis have an earlier onset age than those with only one disease [73]. The present research explored the significance of immune processes and inflammatory responses in the pathogenesis of both CD and psoriasis. These processes encompass granulocyte activation, granulocyte chemotaxis, neutrophil activation and chemotaxis, and lymphocyte differentiation. It has been reported that the common influencing factor of the two is immune factors, and they share many common susceptibility sites, such as 9p24 near JAK2, 10q22 near ZMIZ1 and 11q13 near PRDX5 [74]. Bettelli et al. found that the IL-23/Th17 immune response axis plays an important role in many autoimmune related diseases, including psoriasis and Crohn's disease [75,76]. Certain immune cells within vulnerable populations may generate TNF-α in response to external conditions and infection [77], IL-1β, IFN-α and IL-6, thereby promoting the production of IL-23 by dendritic cells [78]. It is insufficient to stimulate the development of

primitive T cells into Th17 cells [76], but it can promote the expansion and survival ability of Th17 cells, leading to the large release of IL-17 and other cytokines, promoting the aggregation of inflammatory cells, and leading to inflammatory reactions [74]. At the same time, IL-17 can enhance keratinocyte stimulation and proliferation [73], promote the secretion of a large amount of proteases by myofibroblasts in the intestine, and cause tissue damage [79].

This study, through bioinformatics analysis, reveals for the first time that neutrophil infiltration and the S100-chemokine axis constitute a core molecular hub in the comorbidity of the two diseases, complementing previous research. The core genes identified in this study, such as S100A12, S100A9, CXCL8, and CXCL1, are closely related to the regulation of neutrophil function. S100A12 and S100A9 can promote the migration of neutrophils to inflammatory sites by activating pattern recognition receptors (such as TLR4, RAGE) and induce the formation of neutrophil extracellular traps (NETs) [80,81]. NETs have been confirmed to exacerbate tissue damage and autoimmune reactions by releasing histones and antimicrobial proteins in both Crohn's Disease (CD) and psoriasis [82,83]. Moreover, chemokines CXCL8 (IL-8) and CXCL1 are key signaling molecules for neutrophil chemotaxis; their high expression recruits a large number of neutrophils to inflammatory sites and activates the downstream NF-κB pathway by binding to the receptor CXCR2, driving the release of pro-inflammatory factors such as IL-1β and TNF-α [84]. The synergistic action of these genes may form a "neutrophil-chemokine axis," where chemokines recruit and activate neutrophils, which in turn secrete S100A proteins and chemokines, forming a positive feedback loop that continuously amplifies the inflammatory response. This mechanism may lead to comorbid phenotypes through shared immune microenvironment dysregulation in both diseases. This hypothesis is supported by previous studies; for example, the S100A8/A9 complex is significantly elevated in psoriasis serum and positively correlates with disease activity [85], while CXCL8 inhibitors can significantly reduce intestinal inflammation in animal models [86]. However, unlike single-disease studies, our findings show that these molecules exhibit synergistic high expression in comorbidity, suggesting that the "neutrophil-chemokine axis" might be a common pathway across gut and skin barriers. Future research needs to further dissect the interactions of key molecules within this axis and their tissue-specific regulatory mechanisms.

For microRNAs, which mainly play a role in regulating gene expression, they play a very important role in the onset and development of many autoimmune diseases including CD and psoriasis [87,88]. Our bioinformatic analysis of several miRNAs such as hsa-mir-24-3p, hsa-mir-222-3p, hsa-mir-146a-5p and hsa-mir-21-5p may play a critical role in the development of CD and psoriasis, and it is expected to be a new target for CD and psoriasis. In T cells of psoriasis patients, miR-21 is highly expressed. However, after UV B treatment, the levels of miR-21 decreased, indicating that low expression of miR-21 contributes to the treatment of psoriasis. Blocking miR-21 with anti miR-21 oligomers can lead to varying degrees of decrease in epidermal cell thickness in psoriasis. Research has found that Atritin can inhibit the development of psoriasis vulgaris by reducing the levels of mir-21-5p [89]. In Crohn's disease, mir-21-5p has been identified as a potential non-invasive biomarker [90]. MiR-146a-5p shows association with the clinical response of individuals with psoriasis undergoing treatment with the tumor necrosis factor-alpha inhibitor adalimumab [91]. Hsa-circRNA-102685 possibly has a role in CD etiology by activating mir-146a-5p [92].The miR-222-3p activity is enhanced in DSS induced colitis mice [93]. The specific association of miR-222/TIMP3 has the potential to have a function in the etiology and progression of psoriasis [94]. Real time PCR detection of disease-related miRNAs revealed downregulation of miR-24-3p expression in psoriasis patients [95]. In the experimental colitis mouse model, miR-24-3p can promote the polarization of M2 macrophages and alleviate colon damage caused by excessive inflammation [96]. At present, the research of various diseases and miRNAs is a hot spot, and the results we obtained through bioinformatic analysis provide new ideas for the diagnosis and treatment of CD and psoriasis.

These key genes obtained from our study are mainly enriched in inflammatory cell migration and chemotaxis pathways. Several studies have shown a direct relationship between these pathways and the level of activity of both diseases. We obtained methotrexate via drug prediction within our database based on the major pathways and key genes identified in our research. Methotrexate has been studied for several years for its potential efficacy in the care of individuals with both

disorders, and it has been shown to have a definite effect, which verifies the validity of the present study [97]. The emergence of herbal extracts to control disease management has been a hot research topic in recent years, so we predicted the herbal medicines based on the key pathways and key genes obtained from our study through the database, with a view to providing ideas for the future development of medicines for the treatment of the two diseases.

Despite the achievements of our study, several limitations need to be addressed. Firstly, although we utilized datasets from existing public databases for preliminary validation, we lacked an independent cohort specifically designed for validating the comorbidity of these two diseases. This means our findings may not fully reflect the complexity and diversity in real-world scenarios. Secondly, without distinguishing between potential differences among different subtypes, our results may not be universally applicable. For example, in Crohn's disease, varying clinical presentations and pathological features could influence gene expression patterns [98]. Lastly, while we have confirmed the functions of some key genes, how these genes specifically impact immune processes requires further exploration. To overcome these limitations and advance the field, future research should focus on several directions: firstly, establishing and analyzing more data sets derived from actual patient samples, particularly those clearly labeled as being in a state of comorbidity between Crohn's disease and psoriasis; secondly, refining the scope of the study by considering differences among subtypes and exploring how these differences affect treatment responses; thirdly, integrating experimental biology methods such as cell experiments and animal models to validate bioinformatics predictions, thereby providing a more solid evidence base.

In summary, this study, through innovative data-driven methodologies, fills a critical gap in the understanding of the comorbidity mechanisms between Crohn's Disease (CD) and psoriasis. Its conclusions not only deepen our understanding of the common pathological bases of inflammatory diseases but also provide theoretical support for the development of cross-disease therapeutic strategies. This research highlights the unique advantages of bioinformatics in unraveling the complex mechanisms underlying multifaceted diseases. By identifying key molecular hubs such as the neutrophil infiltration and the S100-chemokine axis, this work underscores the potential for developing novel treatments that could address both conditions simultaneously, thereby opening new avenues for therapeutic interventions across different diseases sharing similar immune dysregulation pathways.

## 5 Conclusions

In conclusion, this large-scale study used integrated computational analysis to uncover critical hub genes (S100A12, CXCL8, IL1RN, S100A9, CXCL10, MMP1, CXCL1, FPR1, CXCR2, S100A8) linked to Crohn's disease (CD) and psoriasis. Inflammation and immunological modulation were identified as common underlying mechanisms in both CD and Psoriasis, driven by neutrophil infiltration. These hub genes, along with the regulatory molecules and associated signaling pathways, have the potential to be a big breakthrough in the recognition and therapy of CD and Psoriasis patients. However, further in-depth research is needed to properly understand the specific shared pathophysiology of these two disorders.

## Author contributions

**Conceptualization:** Tianqi Liu, Xiaoqing Zhang, Ruiqi Chen, Zhepeng Luo.

**Data curation:** Tianqi Liu, Ruiqi Chen.

**Formal analysis:** Tianqi Liu, Yifan Sun.

**Funding acquisition:** Jiani Wang.

**Investigation:** Tianqi Liu, Yifan Sun.

**Methodology:** Tianqi Liu, Ruijian Zhang.

**Project administration:** Tianqi Liu.

**Resources:** Tianqi Liu, Ruijian Zhang.

**Software:** Tianqi Liu, Xiaoqing Zhang.

**Supervision:** Tianqi Liu, Xiaoqing Zhang, Liwen Zhang, Zhepeng Luo.

**Validation:** Tianqi Liu, Xiaoqing Zhang, Liwen Zhang.

**Visualization:** Tianqi Liu, Xiaoqing Zhang.

**Writing – original draft:** Tianqi Liu, Xiaoqing Zhang.

**Writing – review & editing:** Jiani Wang.

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
