## [Decision Letter · Decision Letter 0]

23 Feb 2025

PONE-D-24-43295Uncovering Common Disease Mechanisms and Critical Biomarkers in Crohn's Disease with Concurrent Psoriasis and Exploring Potential Therapeutic AgentsPLOS ONE

Dear Dr. Wang,

Thank you for submitting your manuscript to PLOS ONE. After careful consideration, we feel that it has merit but does not fully meet PLOS ONE’s publication criteria as it currently stands. Therefore, we invite you to submit a revised version of the manuscript that addresses the points raised during the review process by Apr 09 2025 11:59PM.

We look forward to receiving your revised manuscript.

Kind regards,

Laura Calabrese

Academic Editor

PLOS ONE

Journal Requirements:

“Thanks for the support from the National Natural Science Foundation of China (82000528).”

“This work was supported by the National Natural Science Foundation (82000528).”

 “This work was supported by the National Natural Science Foundation (82000528).”

4. Please note that your Data Availability Statement is currently missing accession number of each dataset. If your manuscript is accepted for publication, you will be asked to provide these details on a very short timeline. We therefore suggest that you provide this information now, though we will not hold up the peer review process if you are unable.

Reviewers' comments:

Reviewer's Responses to Questions

**Comments to the Author**

1. Is the manuscript technically sound, and do the data support the conclusions?

Reviewer #1: Yes

Reviewer #2: Partly

2. Has the statistical analysis been performed appropriately and rigorously? 

Reviewer #1: Yes

Reviewer #2: Yes

3. Have the authors made all data underlying the findings in their manuscript fully available?

Reviewer #1: Yes

Reviewer #2: Yes

4. Is the manuscript presented in an intelligible fashion and written in standard English?

Reviewer #1: Yes

Reviewer #2: Yes

5. Review Comments to the Author

Reviewer #1: In this article, Liu et al reported the important molecules and mechanisms responsible for the concomitance of Crohn's disease and Psoriasis by using quantitative bioinformatics utilizing a publicly available RNA sequencing repository. They found that among the identified common DEGs, 40 genes were downregulated and 37 were upregulated, totaling 77 genes. Crohn's disease and Psoriasis had a higher concentration of pathways associated with inflammation. After validation, functionality of hub genes was confirmed for S100A12, CXCL8, IL1RN, S100A9, CXCL10, MMP1, CXCL1, FPR1, CXCR2, and S100A8. The hub genes showed an increase in expression in response to neutrophil infiltration. The expression of S100A12, CXCL8, IL1RN, S100A9, CXCL10, MMP1, CXCL1, FPR1, CXCR2, and S100A8 was found to be significantly linked to immune processes such as neutrophil activation, neutrophil chemotaxis, and neutrophil migration associated with Crohn's and Psoriasis disease. They concluded that S100A12, CXCL8, IL1RN, S100A9, CXCL10, MMP1, CXCL1, FPR1, CXCR2, and S100A8 as the central genes in the pathogenesis of CD and Psoriasis comorbidity. The significance of neutrophil infiltration in promoting inflammatory and immune-mediated dysfunction seems to be crucial in the etiology of concurrent Crohn's and Psoriasis, offering an avenue for diagnostic and therapeutic methods. These are very interesting research results. The research method used is appropriate, using an exclusive analysis method. The results obtained are reasonable and very interesting. The discussion items are also very interesting data and discussions, although the contents of the discussions are very diverse due to the exclusive analysis. There are no particularly inappropriate parts or additional questions.

Reviewer #2: In the manuscript, the authors reported that Crohn’s disease and psoriasis share significant inflammatory pathways, yet the molecular mechanisms underlying their co-occurrence remain unclear. Using publicly available RNA sequencing data from the Gene Expression Omnibus, the study employed bioinformatics approaches to identify differentially expressed genes (DEGs) and protein-protein interaction (PPI) networks associated with both diseases. The analysis revealed 77 common DEGs, with 40 downregulated and 37 upregulated genes, highlighting key inflammatory pathways. Validation confirmed the central role of hub genes, including S100A12, CXCL8, IL1RN, S100A9, CXCL10, MMP1, CXCL1, FPR1, CXCR2, and S100A8, which were strongly linked to neutrophil activation, chemotaxis, and migration. These findings may imply that neutrophil infiltration plays a pivotal role in the shared pathogenesis of Crohn’s disease and psoriasis, offering potential diagnostic and therapeutic targets. To enhance the manuscript, the following issues need to be addressed appropriately.

Major Comments

• It is unclear whether the psoriasis dataset includes patients with concurrent Crohn’s disease and vice versa. If multiple patients in each dataset have both diseases, this could introduce bias in the hub gene analysis results. Clarification on this point is necessary.

• To enhance clinical validation, the expression levels of the identified hub genes should be confirmed in patients who have both Crohn’s disease and psoriasis.

• While the study mentions a 9.6% prevalence of psoriasis in Crohn’s disease, there is no discussion on the prevalence of Crohn’s disease among psoriasis patients. A bidirectional approach is essential when examining the relationship between these two diseases. Additional discussion in the Introduction and Discussion sections is recommended.

• Psoriasis comprises multiple subtypes, including psoriasis vulgaris, psoriatic arthritis, and pustular psoriasis, each associated with distinct genetic polymorphisms. It is important to clarify which specific subtype(s) are represented in the psoriasis dataset and whether a similar classification exists for Crohn’s disease in the dataset used.

• The identified hub genes are primarily associated with general immune and inflammatory responses. However, there is insufficient discussion on why these specific genes are particularly relevant in the comorbidity of Crohn’s disease and psoriasis. A deeper mechanistic analysis and discussion would strengthen the novelty of the findings.

Minor Comment

• In the Abstract, the abbreviation PPI (Protein-Protein Interaction) appears for the first time without explanation. A brief description should be provided to ensure clarity for readers.

6. PLOS authors have the option to publish the peer review history of their article (what does this mean? ). If published, this will include your full peer review and any attached files.

**Do you want your identity to be public for this peer review?** For information about this choice, including consent withdrawal, please see our Privacy Policy .

Reviewer #1: No

Reviewer #2: No

---

## [Author Response · Author response to Decision Letter 1]

13 Apr 2025

Deer editors and reviewers,

We thank all the editors and reviewers for your valuable comments and suggestions. We have carefully revised the manuscript to enhance its clarity and facilitate the understanding of the readers. Our point-to-point responses are presented in the following. We hope that the revision would satisfactorily address the comments and concerns of the editors and reviewers.

Style requirements

We have revised the style of the manuscript according to the requirements of journal.

Acknowledgments

We have removed the fund support mentioned in the acknowledgements.

Financial disclosure

We have modified it to “This work was supported by the National Natural Science Foundation (82000528). The funders had no role in study design, data collection and analysis, decision to publish, or preparation of the manuscript.”

Data Availability Statement

We have modified it to “This study utilized the following publicly available gene expression datasets: GSE95095, GSE13355, GSE102133, and GSE14905. These datasets can be queried and downloaded from the Gene Expression Omnibus (GEO) database (https://www.ncbi.nlm.nih.gov/geo/) using the accession numbers mentioned above.”

Reviewer #1

Comments:

“These are very interesting research results. The research method used is appropriate, using an exclusive analysis method. The results obtained are reasonable and very interesting. The discussion items are also very interesting data and discussions, although the contents of the discussions are very diverse due to the exclusive analysis. There are no particularly inappropriate parts or additional questions.”

Response:

We thank the reviewer for recognizing the technical validity of our work. All conclusions have been carefully re-examined to ensure they strictly align with the presented data.

Reviewer #2

Major Comments

Comment #1:

“It is unclear whether the psoriasis dataset includes patients with concurrent Crohn’s disease and vice versa. If multiple patients in each dataset have both diseases, this could introduce bias in the hub gene analysis results. Clarification on this point is necessary.”

Response #1:

We confirm that the original datasets (GSE95095 and GSE13355) do not specify comorbid status. This study focuses on shared mechanisms between the two diseases. We have added this limitation in the Discussion and recommended future studies to validate findings in comorbid cohorts (Pages 24-25, Lines 528-545).

Comment #2:

“To enhance clinical validation, the expression levels of the identified hub genes should be confirmed in patients who have both Crohn’s disease and psoriasis.”

Response #2:

We fully agree that clinical validation in comorbid patients would strengthen our findings. However, due to the scarcity of publicly available transcriptomic datasets for CD-PSO comorbidity and ethical barriers to recruiting new comorbid cases within the revision timeline, direct validation was unfeasible. To address this, we:

Validated hub gene diagnostic value in independent CD (GSE102133) and Psoriasis (GSE14905) cohorts.

Added literature evidence linking these genes to neutrophil pathways in both diseases (Pages 22-23, Lines 465-491).

Explicitly stated this limitation and proposed future validation in the Discussion (Pages 24-25, Lines 528-545).

Comment #3:

“While the study mentions a 9.6% prevalence of psoriasis in Crohn’s disease, there is no discussion on the prevalence of Crohn’s disease among psoriasis patients. A bidirectional approach is essential when examining the relationship between these two diseases. Additional discussion in the Introduction and Discussion sections is recommended.”

Response #3:

We have expanded the Introduction with new epidemiological data:

“Patients with psoriasis are 1.70 times more likely to develop CD” (Page 4, Line 77).

Comment #4:

“Psoriasis comprises multiple subtypes, including psoriasis vulgaris, psoriatic arthritis, and pustular psoriasis, each associated with distinct genetic polymorphisms. It is important to clarify which specific subtype(s) are represented in the psoriasis dataset and whether a similar classification exists for Crohn’s disease in the dataset used.”

Response #4:

We carefully reviewed all publicly available datasets of Crohn's disease and psoriasis used in this study. In the description of the data sources, none of the datasets mentioned specific subtypes of Crohn's disease or specific subtypes of psoriasis (such as psoriasis vulgaris, psoriatic arthritis, pustular psoriasis, etc.). We have pointed out the limitations of the paper in our discussion. Suggestions for future research can be further validated through subtype stratification analysis(Pages 24-25, Lines 528-545).

Comment #5:

“The identified hub genes are primarily associated with general immune and inflammatory responses. However, there is insufficient discussion on why these specific genes are particularly relevant in the comorbidity of Crohn’s disease and psoriasis. A deeper mechanistic analysis and discussion would strengthen the novelty of the findings.”

Response #5:

We have added a new mechanistic hypothesis in Discussion(Pages 22-23, Lines 465-491).

Minor Comment

Comment:

“In the Abstract, the abbreviation PPI (Protein-Protein Interaction) appears for the first time without explanation. A brief description should be provided to ensure clarity for readers.”

Response:

Revised to "protein-protein interaction (PPI) network" (Page 2, Line 33).

We tried our best to improve the manuscript and made some changes marked yellow in revised paper which will not influence the content and framework of the paper. We appreciate for editors and reviewers’ warm work earnestly, and hope the correction will meet with approval. Once again, thank you very much for your comments and suggestions. If you need further information or have any other suggestions, please feel free to let us know and we will fully cooperate.

Best regards!

---

## [Editor Report · Decision Letter 1]

20 Apr 2025

Uncovering Common Disease Mechanisms and Critical Biomarkers in Crohn's Disease with Concurrent Psoriasis and Exploring Potential Therapeutic Agents

PONE-D-24-43295R1

Dear Dr. Wang,

We’re pleased to inform you that your manuscript has been now judged scientifically suitable for publication and will be formally accepted for publication once it meets all outstanding technical requirements.

Kind regards,

Laura Calabrese

Academic Editor

PLOS ONE

---

## [Editor Report · Acceptance letter]

PONE-D-24-43295R1

PLOS ONE

Dear Dr. Wang,

I'm pleased to inform you that your manuscript has been deemed suitable for publication in PLOS ONE. Congratulations! Your manuscript is now being handed over to our production team.

Kind regards,

on behalf of

Dr. Laura Calabrese

Academic Editor

PLOS ONE